# Thermal Stability Evaluation of Polystyrene-Mg/Zn/Al LDH Nanocomposites

**DOI:** 10.3390/nano9111528

**Published:** 2019-10-27

**Authors:** Miguel Ángel De la Rosa-Guzmán, Ariel Guzmán-Vargas, Nicolás Cayetano-Castro, José Manuel Del Río, Mónica Corea, María de Jesús Martínez-Ortiz

**Affiliations:** 1Instituto Politécnico Nacional-ESIQIE, Laboratorio de Investigación en Materiales Porosos, Catálisis Ambiental y Química Fina, UPALM, Edificio 7 P. B., Zacatenco, C. P., Ciudad de México 07738, Mexicoaguzmanv@ipn.mx (A.G.-V.); 2Instituto Politécnico Nacional-ESIQIE, Laboratorio de Investigación en Polímeros y Nanomateriales, UPALM, Edificio Z-5, P. B., Zacatenco, Gustavo A. Madero, C. P., Ciudad de México 07738, Mexico; jm.delrio.garcia@gmail.com; 3Instituto Politécnico Nacional, Centro de Nanociencias Micro y Nanotecnologías, C.P., México D.F. 07300, Mexico; ncayetanoc@ipn.mx

**Keywords:** polystyrene, layered double hydroxides, hydrophobization, nanocomposites, degradation temperature, decomposition kinetic

## Abstract

A series of samples of Mg/Zn/Al LDHs (layered double hydroxides) materials was prepared by the co-precipitation and urea hydrolysis methods. They were modified with organic surfactants (acrylate and oleate anions) and characterized by X-ray diffraction, which corroborated the intercalation of anionic species into the interlayer space. The hydrophobized materials were incorporated at low contents (10 and 15 wt.%) to polystyrene, which was synthesized by emulsion polymerization techniques. The polymeric composites were analyzed by thermogravimetry to determine the decomposition temperature. The results demonstrated that the materials with Zn presented the greatest increment in the degradation temperature (7 °C < T < 54 °C). Moreover, the Friedman, Flynn–Wall–Ozawa, and Coats–Redfern models were compared to obtain the kinetic parameters of degradation process. The obtained order of decomposition of the Coats–Redfern model showed that the decomposition process occurs in at least two stages. Finally, the addition of environmentally friendly modified Layered Double Hydroxides (LDH) nanomaterials to the polystyrene (PS) matrix allowed for obtaining polymeric composites with higher thermal stability, retarding the decomposition process of PS.

## 1. Introduction

Flame retardants (FRs) are additives that have been incorporated into polymers to reduce their flammability [1]. FRs are used to minimize ignition, pyrolysis, and combustion in polymeric matrices in order to enhance the properties of composite materials [2,3,4]. The main chemical groups used as FRs are organic-halogen compounds, organic-phosphorus compounds, metal hydroxides, and cationic clays [4,5,6]. Additionally, a good dispersion of flame retardants within a polymeric matrix is an important parameter for increasing the decomposition temperature, modifying the ignition point, and providing greater thermal stability [7,8,9].

This can be accomplished by the organophilization of the inorganic nano-additive, which forms a barrier that minimizes the mass transfer that is associated with polymer degradation [1,10]. In addition, such barriers also serve to reduce the peak in the heat release rate [11]. Consequently, the thermal stability of the inorganic additive is an important factor to consider in the structure and morphology of composite materials [12,13].

On the other hand, Layered Double Hydroxides (LDH) have been used as additives in organic polymers to increase the mechanical, thermal, flame retardant, and barrier properties, as well as to improve their strength [6,14,15]. LDH materials, which have also referred as anionic clays or hydrotalcite like materials, are anion- exchangeable layered compounds: divalent (*M^II^*) and trivalent (*M^III^*) cations are bonded octahedrally by OH^−^ groups to form brucite-like layers [16,17]. Their general formula is: ${\left[M_{1-x}^{II}M_{x}^{III}{\left(OH\right)}_{2}\right]}^{x}\left[A_{x/n}^{n-}\right]\cdot mH_{2}O$, where *A^n−^* represents the compensating negatively-charged species, located within the interlayer space [18,19,20].

In recent researches, the incorporation of hydrotalcite type materials has had different applications, such as pharmaceuticals, adsorbents, catalysts, as well as additives in polymeric materials [5,21].

Anionic clays have tunable properties and the possibility of anion exchange. These characteristics allow for the development of materials with hydrophobic surfaces, which is necessary to achieve a good interaction with organic materials, such as polymeric matrices [10,22]. However, the presence of inorganic anions and water in the interlayer space of the structure makes the incorporation of organic anions in LDH materials that are desirable to improve the dispersion into polymeric materials difficult and it could cause the formation of particle aggregates in the matrix [6,23]. The incorporation of LDHs increases the degradation temperature of nanocomposites and influences their degradation rate [24].

Some works have reported the incorporation of LDH (Zn-Al, Co-Al, and Mg-Al) in polystyrene matrixes in order to be used as flame retardants and thermal stabilizers. Their results showed an increment of thermal stability close to 18 °C when LDH are incorporated at 5 wt.% and modified with dodecyl sulfate anion [9].

It has been reported that LDH materials, as flame retardant, present three characteristics through the thermal decomposition as heat absorption, gas dilution, and ash formation [5]. In addition, during this process, the loss of interlayer water in LDH, as well as intercalated anions and the water generated by the dehydroxylation, allows for the reduction of the available fuel. For this reason, the mass loss rate is significantly reduced. In this work, a variety of multicationic LDHs (containing Mg, Zn, and Al) were synthesized by the co-precipitation and urea hydrolysis methods. These materials were modified with oleate and acrylate anions, so that they could be dispersed into polystyrene (PS) matrix, because they are short-chain organic compounds and they have a good hydrophobic-lipophilic balance (HLB ≈ 1); to be precise, the lipophilic part has almost the same size of the hydrophilic part. This characteristic allows for the inorganic materials, such as LDH, to be hydrophobized [25,26] Thus, the incorporation of modified LDH materials was carried out and their thermal stability was studied in order to increase the thermal stability of the PS matrix, generating environmentally friendly composite materials.

## 2. Materials and Methods

### 2.1. Materials

Magnesium nitrate hexahydrate (Mg(NO_3_)_2_·6H_2_O ≥ 98%), zinc nitrate hexahydrate (Zn(NO_3_)_2_·6H_2_O ≥ 98%), aluminum nitrate nonahydrate (Al(NO_3_)_3_·9H_2_O ≥ 98%), sodium hydroxide (NaOH ≥ 98%), acrylic acid (C_3_H_4_O_2_
≥ 99%), oleic acid (C_18_H_34_O_2_
≥ 99%), styrene (C_8_H_8_
≥ 99%), sodium persulfate (Na_2_S_2_O_8_
≥ 98%), and toluene (C_8_H_8_
≥ 99.5%) were all obtained from Sigma-Aldrich (Saint Louis, MO 63103, USA). Additionally, surfactant ABEX EP-100 ≥ 35% (SOLVEY Toronto, ON, M5J2T-3, CA) and distilled water were also used as received. 

### 2.2. LDH Synthesis

Mg/Zn/Al LDH materials with a constant *M^II^/M^III^* mole ratio of 3 were prepared by co-precipitation at constant pH and by the urea hydrolysis methods while using Zn(NO_3_)_2_·6H_2_O, Mg(NO_3_)_2_·6H_2_O, and Al(NO_3_)_3_·9H_2_O as precursors. 

The use of two different methods of synthesis was established to obtain pure phases of the hydrotalcite like-materials. In the case of materials with a higher content of Zn, they must be synthesized by the urea hydrolysis method with the aim of avoiding the formation of the additional phase of ZnO, which is usually obtained when materials are synthetized by coprecipitation method [27,28,29]. Thus, the prepared samples were labeled according to their Mg and Zn content, Table 1, the synthesis methods, their respective theoretical molar ratio, and the last column presents the anionic exchange capacity (AEC) of the samples (mmol per gram of LDH), which is used to determine the amounts of organic anions that were intercalated in the these LDH samples.

Figure 1 shows a scheme of the LDH structure. It should be noted that Figure 1 also illustrate an octahedral unit to specify that, for each sample, the content of divalent cations (Mg^2+^ and/or Zn^2+^) and the trivalent cation (Al^3+^) were varied.

#### 2.2.1. Co-Precipitation at Constant pH

The LDH samples were prepared at constant pH, with a 2 M NaOH solution. The pH was maintained constant while using a pH-Stat Titrando (Metrohm, Switzerland), as previously reported [30]. The obtained suspension was aged overnight at 80 °C for 13 h. The solids were washed with distilled water and dried at 80 °C for 15 h.

#### 2.2.2. Urea Hydrolysis

Based on the method used by Alexandra Inayat et al. (2011) [31], metal nitrate salts were used as precursors, and they were mixed with urea/NH_4_NO_3_ in a three-neck flask. The suspension was stirred at 90 °C under a N_2_ atmosphere for 12 h. The obtained solid was washed with distilled water and then dried at 80 °C for 15 h. 

### 2.3. Thermal Treatments

Synthesized materials by the co-precipitation method were calcined in air flow at 450 °C for 3 h to obtain the corresponding mixed oxides. In the case of samples that were prepared by urea hydrolysis, they were thermally treated in N_2_ flow at 450 °C for 3 h. Again, the control of the thermal treatment atmosphere for samples that were synthetized by urea hydrolysis method prevents the formation of ZnO as additional phase [29,32,33].

### 2.4. Modification of LDHs with Anionic Surfactants

LDH samples were modified with the anionic species of short chain organic compounds: acrylic acid (AAC) and oleic acid (AOL), in order to increase their compatibility with the polymeric matrix.

#### 2.4.1. Acrylate Anion

Calcined LDH materials were added to an aqueous solution of acrylic acid, while using a molar ration of 1:1.25 according to their anion exchange capacity (AEC). The mixture was stirred at room temperature under N_2_ for seven days.

#### 2.4.2. Oleate Anion

After thermal treatment, mixed oxides and oleic acid were added to a 50/50 *v*/*v* methanol/water solution, while using a molar ratio of 1:1.25 according to their anion exchange capacity (AEC). The mixtures were stirred for seven days at room temperature under N_2_.

### 2.5. Polystyrene Synthesis

Polystyrene was synthesized by emulsion polymerization to yield 200 g of latex containing 60 wt.% of solid and Table 2 provides the recipe details.

Polymerization was carried out at 75 °C under N_2_ while using a semi-continuous process in a 0.5 L jacketed glass reactor and a feeding tank. A dosing pump ensured a continuous flow of pre-emulsion material. The stirring speed and addition rate were controlled at 250 rpm and 1.3 g·min^−1^, respectively.

### 2.6. Nanocomposite Preparation

Dispersions of modified LDHs in toluene were stirred by ultrasound and then incorporated into the polystyrene matrix at different concentrations (10 and 15 wt.%). The mixtures were stirred for 24 h and, finally, solvent was removed and recuperated with a rotavapor at room temperature.

### 2.7. Characterization and Measurements

Powder X-ray diffraction (XRD) patterns were obtained with a Philips PRO instrument (Malvern, United Kingdom) while using Cu-Kα radiation (1.5406 Å, 35 kV, 25 mA). Materials were also characterized using Fourier transform infrared spectroscopy with attenuated total reflectance (FTIR-ATR, Perkin Elmer, FRONTIER, Waltham, MA USA) and MALDI-TOF ionization, Matrix-assisted laser desorption/ionization time-of-flight mass spectrometer, (AUTOFLEX MALDI-TOF, Bruker, DALTONIC, Billerica, MA USA). The thermal behavior of the nanocomposites from 30 to 720 °C was evaluated while using TGA, Thermogravimetric Analyzer and dTG analysis, derivate of the TG, (Perkin Elmer, STA 6000, Waltham, MA USA) with N_2_ flow of 20 mL·min^−1^ and scan rates of 5, 10, 20, and 30 °C·min^−1^. The microstructural analysis of polymeric composites was performed with a scanning electron microscope (SEM) (JEOL JSM6701F, Tokyo, Japan), working at 5.0 kV. Further, Au-Pd coating was made on the samples to increase the conductivity. The morphology of composites was also studied by transmission electron microscopy (TEM) (JEOL JEM-2100, Tokyo, Japan) working at 200 kV. The instrumental magnification ranged at 2 × 10^5^.

## 3. Results and Discussion

### 3.1. XRD Results

The crystallographic structures of fresh and calcined samples were identified while using powder X-ray diffraction (Figure 2).

Freshly-prepared LDH materials showed the structure corresponding to the natural mineral hydrotalcite (JCPDS: 22-0700 [34,35,36], Figure 2a). After thermal treatment (Figure 2b), the diffraction patterns showed characteristic reflections that corresponded to the periclase phase (JCPDS: 00-042-1022), and for samples containing Zn, zincite phase was also observed (ZnO) (JCPDS: 00-036-1451) [37,38,39]. Figure 3 presents the obtained diffractograms for the materials after anion exchange with acrylic or oleic acids.

The results showed that the LDH materials treated with acrylate and oleate species lost crystallinity with respect to the original samples. However, the diffractograms (Figure 3) present the characteristic peaks of the hydrotalcite phase. In the case of samples with Zn, an additional peak, corresponding ZnO phase is observed, except for the sample Mg2ZnAl-AAC, where a pure hydrotalcite like material is obtained and its crystalline degree is higher when compared to the rest of materials [37,38,40].

It is also important to remark that all of the samples showed a shift toward smaller angles in the (003) plane, which is attributed to the incorporation of the organic anions (acrylate (AAC) and oleate (AOL)) into the interlayer space. Table 3 shows the cell parameters of fresh and modified LDH.

When considering the sizes of the anionic species intercalated into the layer space of the LDH materials, the changes in the cell parameters shown in Table 3 can be explained with the schemes of Figure 4. For the acrylate anion, it is proposed that its incorporation into the interlayer space occurs in pairs; this is suggested by the increment of d_003_ distance. For example, the d_003_ distance is 13.3 Å for 2MgZnAl-AAC material. This value resulted in the thickness of an LDH sheet (4.8 Å) and the incorporation of two acrylate anions in interaction with the material sheets, where the average length of each anion is 6 Å and the thickness is 2.8 Å, giving a distance that is occupied by the anions of 8.5 Å. This value is smaller than the sum of two acrylate species, because the anion is not incorporated in a parallel way (Figure 4a) [26,41,42]. On the other hand, the oleate anion has a length close to 50 Å (in its extended form) and a thickness of 6 Å. In this case, it is assumed that oleate is inserted into the interlayer space parallel to the LDH sheets (Figure 4b) [26]. Following the above example, the obtained distance for 2MgZnAl-AOL is 15.5 Å, attributed to the contribution of oleate as compensation anion.

### 3.2. FTIR Spectroscopy

Fresh samples, modified LDH, and nanocomposites were analyzed by FT-IR spectroscopy. The spectra present similar bands for each family of samples. Figure 5 shows the FTIR spectra for all MgAl materials as an example.

The FTIR spectrum (Figure 5a) for the fresh MgAl LDH presents a band around 3400–3600 cm^−1^, which corresponded to the stretching vibration of the OH^−^ functional group. The vibration around 1640 cm^−1^ is attributed to hydrogen bonding due to hydration, as well as the remaining carbonates that are found as compensating anions in the interlaminar space. Finally, around 1380 cm^−1^, the band corresponding to NO_3_^−^ compensation anions was observed [30].

Modified MgAl LDH (Figure 5b,c) presented an additional vibration at 3100 cm^−1^, corresponding to -CH_2_- bands, and the band between 2850 and 2950 cm^−1^ is associated to the saturation vibration of -CH_3_ groups in the oleate anion, while, for acrylate-modified samples, these bands are smaller and less defined. The vibration corresponding to C=O of the carboxylate group appears around 1605 cm^−1^; the -C=C- double bond was located at 1492 cm^−1^; while, the C-H deformation vibrations are located at 1400 cm^−1^.

The spectra of the nanocomposite materials (Figure 5d,e) showed the characteristic polystyrene bands: the stretching vibration of -CH_2_ at 3000–2700 cm^−1^; the overtones of aromatic groups bonded to the main polystyrene chain at 1600 cm^−1^. A characteristic band of the -C=C- group close to 1750 cm^−1^ would indicate the presence of residual monomer, but it was absent from all spectra (Figure 5). The presence of -CH_3_ and -CH_2_- moieties was consistent with the spectral features between 1750 and 1450 cm^−1^. 

### 3.3. MALDI-TOF Mass Spectrometry

The molecular weight distribution was obtained by MALDI-TOF mass spectrometry by duplicate, and the average molecular weight and polydispersity index (*PDI*) were then calculated (Table 4).

The polydispersity index (PDI) establishes the relationship between the average molar mass and the number of polymer chains [43,44]. The results show PDI that is close to 1; this means the length of the chains of the polystyrene have similar size [45,46].

### 3.4. Scanning Electron Microscopy (SEM)

A key factor in obtaining the nanocomposites is the interaction of the polymeric matrix with the organic clay [47]. Thus, SEM characterized composite materials. Figure 6 presents the micrographs of PS/2MgZnAl-AAC (15 wt.%) and PS/Mg2ZnAl-AAC (15 wt.%) as examples. Figure 6a) shows a porous surface with heterogeneous sizes. Segregated additive particles were observed in the border of the pores, which indicates incomplete incorporation of them into the polymeric matrix. Figure 6b) presents a dense and homogeneous surface with less porosity. This suggests that modified LDH were better dispersed into the PS, generating higher compatibility in the final composite material.

### 3.5. TEM

For a better understanding of SEM results, the corresponding samples were analyzed by TEM. PS/2MgZnAl-AAC (15 wt.%) sample is shown in Figure 7a,b, and the images show that effectively, heterogeneous size aggregates of LDH particles are observed in the border of the material pores. One can also observe aggregates of 20, 11, 7, and 3 nm on the film surface. On the other side, PS/Mg2ZnAl-AAC (15 wt.%) sample, Figure 7c,d, present homogeneous and well dispersed LDH particles on the film surface with an average size from 1.3 to 1.7 nm.

### 3.6. Thermal Analysis

Modified LDH with AAC and AOL were analyzed by thermogravimetry (Figure 8). The curves show a first weight loss at ≅180 °C, which is attributed to physisorbed and interlaminar water loss. After that, at 200 < *T*/°C < 450, the dehydroxylation of inorganic material sheets, the decomposition of the intercalated organic anions (acrylate and oleate), as well as the nitrates and CO_2_ release are observed. Above 450 °C, the material decomposes until the formation of mixed oxides. Finally, the last stage of decompositions process happens up to 700 °C when the spinel phases are formed [48,49,50]. In all thermograms, it is observed that there was a greater weight loss for samples containing AOL; this is normal, since it is a larger anionic species when compared to AAC. In addition, weight loss decreases as Zn content does and this is a consequence of the difference in the anionic exchange capacity (see Table 1).

Figure 9 shows thermograms (and their derivatives) for polystyrene (PS) and nanocomposites with 10 wt.% LDH. The samples showed a mass loss at 150 °C, which corresponded to the gradual loss of water adsorbed in the inorganic materials. Some works have reported that physisorbed water loss happens between 30 to 180 °C [51,52]. The main mass loss occurs over the range 300 < *T*/°C < 450, corresponding to the decomposition of the AAC and AOL anionic species and the polymeric chains, and dehydroxylation of LDH layers. Above 420 °C, the total loss of polymer occurs and mixed oxides are formed [53]. Indeed, it is noted that, at higher temperatures, there is not residues after total thermal degradation of the polymer.

Figure 10 shows thermograms (and their derivatives) for PS and nanocomposites with 15 wt.% LDH. The latter present similar thermal decomposition profiles to the composite materials presented in Figure 9. However, the first mass loss at around 150 °C is not clearly observed in the nanocomposites because a gradual decomposition occurs. This can be attributed to LDH materials being better incorporated into the polymeric matrix, causing the water to gradually desorb during all decomposition processes and not suddenly, as occurs in samples with 10 wt.% of LDH [54,55].

Table 5 summarizes the decomposition temperatures obtained from the thermograms. 

The depolymerization of the samples PS/MgAl-AOL (10 wt.%), PS/2MgZnAl-AAC, PS/2MgZnAl-AOL, and PS/Mg2ZnAl-AAC (15 wt.%) occurs at higher temperatures than neat PS (Table 5) [56]. This increment of decomposition temperature shows that the incorporation of modified LDH improves the thermal properties of the polystyrene matrix by slowing its decomposition, which reduces the mobility of the polymeric chains and therefore retards the degradation of PS [56,57].

### 3.7. Decomposition Kinetic Analysis

The decomposition kinetics were obtained from the thermogravimetric data by fitting them while using several models. This analysis was performed on all samples, but only the results for PS/Mg2ZnAl-AAC (15 wt.%) are presented as example, because of its remarkable value of decomposition temperature (445.85 °C). 

In general, it has been reported that the thermal decomposition of polymeric matrices with inorganic additives occurs by the breaking of the macromolecular chains [58], and it typically follows the pattern:Polymer→volatiles+ash.

The conversion degree (α) is defined as Equation (1):(1)$$\alpha =\frac{w_{0}-w_{n}}{w_{0}-w_{\infty }}\times 100\%$$
where $w_{n}$ is mass at time *n*, $w_{0}$ is the initial mass, and $w_{\infty }$ is the mass at the end of the thermogravimetric analysis. The kinetics of decomposition can be represented according to Equation (2) [58,59,60,61].
(2)$$\frac{d\alpha }{dt}=kf\left(\alpha \right)$$
and the decomposition constant can be determined by the Arrhenius Equation (3):(3)$$k=A\;\exp\left(-\frac{E}{RT}\right)$$
where *A* represents the pre-exponential factor, *E* denotes the activation energy, *R* is the ideal gas constant ($8.314\;J\cdot mol^{-1}\cdot K^{-1}{)}^{\;}$ and *T* is the temperature in Kelvin. Experiments are performed using a constant heating ramp (β) as shown in Equation (4) [58,60].
(4)$$\frac{d\alpha }{dt}=\frac{A}{\beta }\exp\left(-\frac{E}{RT}\right)f\left(\alpha \right)$$

All kinetic models of decomposition establish the kinetic parameters via Equation (4).

#### 3.7.1. Friedman Method

The Friedman kinetic model works in the conversion range (*α*) for which $\frac{d\alpha }{dT}$ is linear, as shown in Equation (5) [62]:(5)$$\ln\left(\frac{d\alpha }{dT}\right)=f\left(\frac{1}{T}\right)$$

The data that were obtained from the thermogravimetric analysis of the sample PS/Mg2ZnAl-AAC 15 wt.% were plotted as $\frac{d\alpha }{dT}$ against *α* at several β(10, 20 and 30 °C/min). This enabled the line adjustment at the beginning of the thermal decomposition (Figure 11) to be established. The data that were made at 5 °C/min are not reported because they did not present lineal behavior for the Friedman model, which causes high uncertainty.

The linear part of the Friedman model was obtained at 0.1 < *α* < 0.6 and fitted to Equation (6) [59].
(6)$${\left[\ln\left(\beta \cdot \frac{d\alpha }{dT}\right)\right]}_{y}=\ln\;\left(f\left(\alpha \right)\cdot A\right)-\frac{E_{a}}{R}\cdot {\left[\frac{1}{T_{\alpha }}\right]}_{x}$$

Figure 12 shows the obtained results, from the fit of (ln (d*α/*d*T*)) as function of 1/*T* for the sample PS/Mg2ZnAl-AAC 15 wt.% at several heating rates. The slopes of the straight lines (−*E_a_*/*R*) were obtained by linear regression.

Table 6 sows the activation energies for the obtained linear range via the Friedman method for all composites for each heating rate.

The data from Table 6 show an increase of the activation energy (a stabilizing effect) of the PS matrix by the presence of the LDH materials in the composite materials [63]. It is reasonable to propose that the presence of modified LDH dispersed in the matrix generates a barrier that prevents oxygen diffusion to the polymer and thus hinders the degradation of the PS-LDH nanocomposites [55,64].

Thus, the activation energy variations indicate that thermal decomposition of the nanocomposites depends on good dispersion of the additive in the polymeric matrix, as well as the concentration of additive used in the formulation [65].

#### 3.7.2. Flynn–Wall–Ozawa Method

Some authors consider that the Friedman model has an erroneous estimation of the activation energy, because it only takes the linear interval of decomposition reaction into account [66,67,68]. For this reason, the thermodynamic data were tested in the Flynn–Wall–Ozawa (FWO) method, which allows for the activation energy to be obtained from advanced degree data as function of temperature. The main advantage of this method is that it does not assume a decomposition mechanism [66]. This model makes use of the Doyle approximation [58], yielding Equation (7), which involves *α* at different heating rates (*β*) [58,66].
(7)$${\left[\log\beta \right]}_{y}=\log\frac{AE_{a}}{g\left(\alpha \right)R}-2.315-\frac{0.457E_{a}}{R}{\left[\frac{1}{T_{\alpha }}\right]}_{x}$$

This model was applied to PS/Mg2ZnAl-AAC (15 wt.%) as an example, because this nanocomposite showed the greatest increment of decomposition at 0.1 < *α* < 0.6. Figure 13 summarizes the results, which plots log (*β*) as function of 1/*T,* obtained at 5, 10, 20, and 30 °C/min heating rates. The slope of the straight-line fits (−0.457 *E_a_*/*R*) provides the activation energy.

The calculated activation energy for all composite materials by means of the FWO method shows two ranges of activation energies. The first corresponds to 0.1 < *α* < 0.3 (390–425) °C and the second to 0.4 < *α* < 0.6 (433–445) °C (Table 7). It has been reported that this behavior occurs when *E*_a_ is dependent of conversion, because the degradation process takes place in multi-steps, that is, where the mechanism of thermal decomposition is not known or complex [69,70]. Some works report that the first stage (0.1 < *α* < 0.3) is associated with the breaking of weak bonds. For polystyrene, it corresponds to the breaking of –CH_2_– bonds in the polymeric chain [69,71]. At the second stage (0.4 < *α* < 0.6), the weak bonds have been consumed and the complete degradation of the material occurs, which breaks the aromatic structures (–C=C–) [71].

These results suggest that the LDH materials have a fire-retardant effect on the present composites, mainly in the second stage of the decomposition process.

The Friedman and FWO methods are considered as complementary models and it can be assumed that the average activation energy (*Ea*-M) from the two models (Table 8) is a reasonable estimate of the overall activation energy for decomposition [66].

The obtained average activation energies for the two models are higher for the composite materials than for polystyrene. The increment of the decomposition temperature is generated thanks to the addition of modified LDH, as these materials release water vapor (interlayer water) and this decreases the oxygen concentration on the surface of the composites, which protects the polymeric chains before the total decomposition of inorganic layered structures takes place [55].

Furthermore, the formation of MgO and Al_2_O_3_ during the calcination of the inorganic sheets allows for them to absorb the acidic combustion gases and releases more water vapor due to the decomposition of hydroxyl groups [55]. Moreover, the formation of ZnO as a decomposition product generates denser residues, which is more environmentally friendly [72,73].

Some of the activation energies that were obtained by the Friedman and FWO models are substantially different. This behavior is thought to arise from the dispersion of the LDH materials in the matrix. They help to form a barrier effect trapping the radical species that formed during the decomposition process, causing these species to perhaps react between themselves avoiding or slowing down the total decomposition of polymer matrix [11,69,74].

It is important to mention that the *E_a_*-M for samples with 15 wt.% of LDH present a higher standard deviation than those with 10 wt.% LDH content. 

#### 3.7.3. Coats–Redfern Method

The thermogravimetry data were tested in the Coats–Redfern (CR) method, because it considers the mechanism of thermal decomposition. This is an integral version of the Arrhenius method that was employed for the determination of kinetic parameters in the thermal decomposition processes (Equation (8)) [70,75].
(8)$${\left[\ln\left(\frac{g\left(\alpha \right)}{T^{2}}\right)\right]}_{y}=\ln\left(\frac{AR}{\beta E}\right)-{\left[\frac{E}{RT}\right]}_{x}$$

In this case, g(α) is an integral function that allows for the thermal decomposition process to be described by different reaction models in the solid phase and considers the order of reaction (n). The function g(α) can be represented according to Equation (9) or (10), depending on the reaction order (*n*) [71,76,77].
(9)$${\left[\ln\left(\frac{1-{\left(1-\alpha \right)}^{1-n}}{\left(1-n\right)T^{2}}\right)\right]}_{y}=\ln\left(\frac{AR}{\beta E}\right)-{\left[\frac{E}{RT}\right]}_{x}\;\mathrm{for}\;\left(n\neq 1\right)$$
(10)$${\left[ln\left(\frac{-ln\left(1-\alpha \right)}{T^{2}}\right)\right]}_{y}=ln\left(\frac{AR}{\beta E}\right)-{\left[\frac{E}{RT}\right]}_{x}\;\mathrm{for}\;\left(n=1\right)$$

Using Equation (8), the activation energy, pre-exponential factor, and *n* can be obtained. Figure 13 shows a plot of $ln\left(\frac{g\left(\alpha \right)}{T^{2}}\right)$ as a function of 1/*T*, for PS/MgAl (10 wt.%), PS/2MgZnAl-AAC (15 wt.%), PS/2MgZnAl-AOL (15 wt.%), and PS/Mg2ZnAl-AAC (15 wt.%). These samples were analyzed by this method, because they presented the highest increment in the decomposition temperature with respect to the polystyrene matrix. 

The obtained results by CR confirm those that were calculated by the Friedman and FWO methods, but with a clearer tendency of the existence of a multi-stage decomposition process [71,78].

All of the materials of Figure 14 exhibited two zones that were attributed to two different decomposition processes. In the first stage, the activation energy is greater as a result of the presence of hydrotalcite-type materials [8,79] (see Table 9). When these materials are used as additives, they generate an endothermic process that can absorb the heat supplied [8,79]. With increasing temperature, the LDH materials begin to release water. This causes a cooling in the polymeric matrix and the produced gases during the pyrolysis process are encapsulated [8]. Finally, mixed oxides (that are produced by the thermal decomposition of LDH materials) cover the polymeric matrix that is similar to a sheet that blocks the heat flow [8,79].

Once the most unstable structures have been degraded, there is a continuous increment of the heat flow and the second stage starts. This implies that an exothermic process begins, causing the decomposition of more stable organic structures, such as aromatic C bonds. This process requires a lower activation energy, because there is an excess of energy coming from the first stage of pyrolysis (see Table 9) [80].

Table 9 shows the obtained reaction orders and it is observed that, in all cases, is 1/3. This is consistent with the “contracting cylinder” mechanism of polymer decomposition processes [76,77], because Coats and Redfern refer that the reaction rate is determined by several steps, which might be due to the diffusion of the gaseous products out of the solid, or the transport of a particular ion, or the breakage of structures bonds [24]. In this work, in addition to the phenomena that were described by Coats and Redfern, the recombination of free radicals from PS could be possible during the decomposition process due to the presence of LDH materials that were modified with organic short chain compounds. Table 9 also shows the parameters of activation energy, frequency factor, and reaction order.

## 4. Conclusions

Mg/Zn/Al LDH nanomaterials were prepared by coprecipitation and urea hydrolysis and their structures were hydrophobized with the acrylate and oleate anions. By XRD analysis, it was observed that hydrotalcite like material phase was obtained for all fresh samples. After the intercalation of acrylate and oleate anions, besides the main hydrotalcite phase, an additional ZnO phase was also observed in the synthetized samples, except for Mg2ZnAl-AAC. The characteristic bands for fresh and intercalated LDH samples were observed by infrared spectroscopy. 

PS was synthetized by emulsion polymerization with an average molecular weight of 1526.9 Da. After that, the composites materials were prepared with 10 and 15 wt.% of modified LDH with the aim of increasing the decomposition temperature of PS and they were characterized by TGA, where the materials containing Zn presented the best results of thermal stability with an increasing of 7 < T/°C < 54 °C with respect to PS (391.9 °C).

According to the XRD results, most of the modified materials with Zn presented an additional ZnO phase, which could cause a part of the additive to not be hydrophobized, which decreases the LDH dispersion into the polymer matrix and, therefore, the thermal stability. The material PS/Mg2ZnAl-AAC (15 wt.%) with the best thermal stability did not present the additional ZnO phase.

Using the TGA data, the decomposition kinetic was calculated while using Friedman, Flynn–Wall–Ozawa and Coats–Redfern methods in order to obtain the activation energy (*Ea*). The obtained *Ea* by the Friedman and Flynn–Wall–Ozawa methods presents different values attributed to a two stages decomposition process. This was confirmed by the results of the Coats–Redfern method, where, in the first stage, *Ea* was higher than the second stage while assuming that this behavior was due to the presence of the modified LDH into the polymeric matrix. From this model, a reaction order of 1/3 was obtained; this indicates that, during the decomposition process, a recombination of free radicals takes place due to the addition of LDH materials modified with organic short chain compounds.

In this work, it was demonstrated that, while using low contents (10 and 15 wt.%) of hydrophobized LDH, it was possible to obtain polymeric composites with a higher thermal stability than neat PS. Finally, these materials have great potential to be applied as flame retardants.

## Figures and Tables

**Figure 1 nanomaterials-09-01528-f001:**
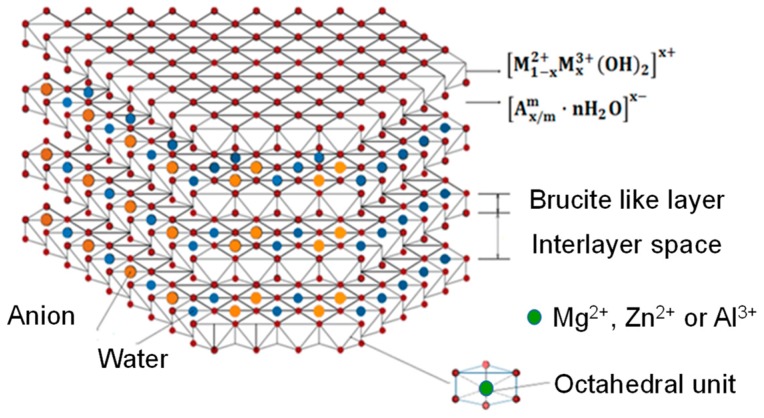
Scheme of the LDH structure.

**Figure 2 nanomaterials-09-01528-f002:**
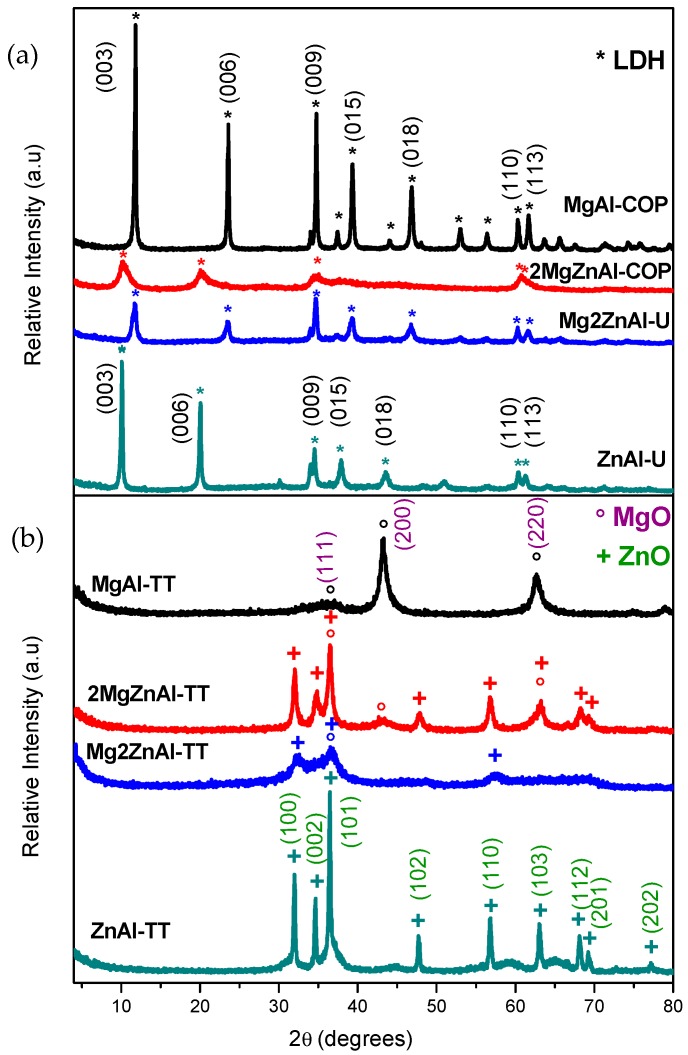
X-ray diffraction patterns of: (**a**) as-prepared, and (**b**) calcined samples.

**Figure 3 nanomaterials-09-01528-f003:**
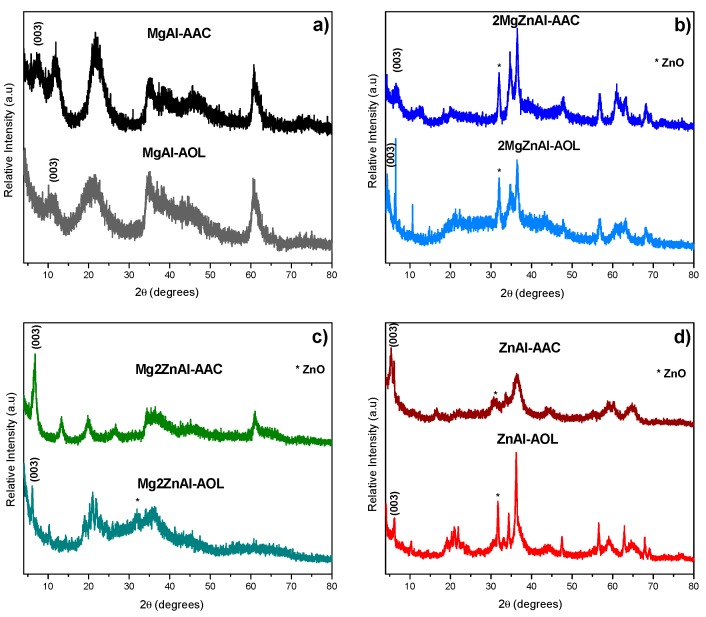
X-ray diffraction patterns of LDHs modified with acrylate (ACC) and oleate (AOL) for: (**a**) MgAl, (**b**) 2MgZnAl, (**c**) Mg2ZnAl, and (**d**) ZnAl.

**Figure 4 nanomaterials-09-01528-f004:**
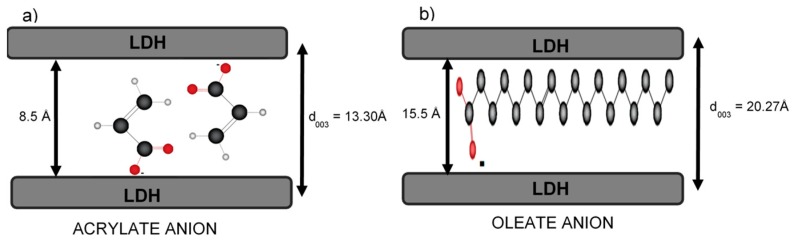
Intercalation scheme of: (**a**) acrylate and (**b**) oleate anions in LDH materials.

**Figure 5 nanomaterials-09-01528-f005:**
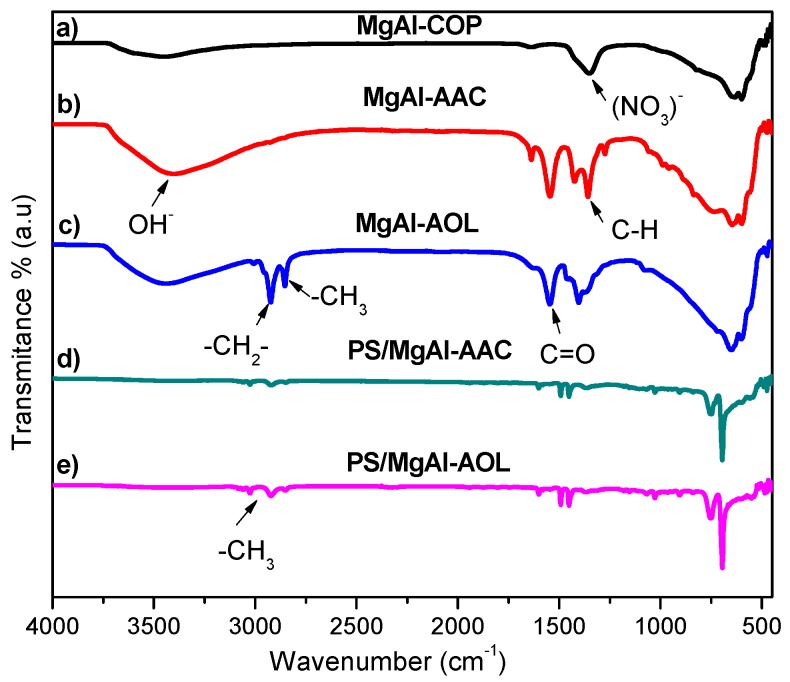
Fourier transform infrared spectroscopy (FTIR) spectra of: (**a**) MgAl-COP, (**b**) MgAl-AAC, (**c**) MgAl-AOL, (**d**) PS/MgAl-AAC, and (**e**) PS/MgAl-AOL.

**Figure 6 nanomaterials-09-01528-f006:**
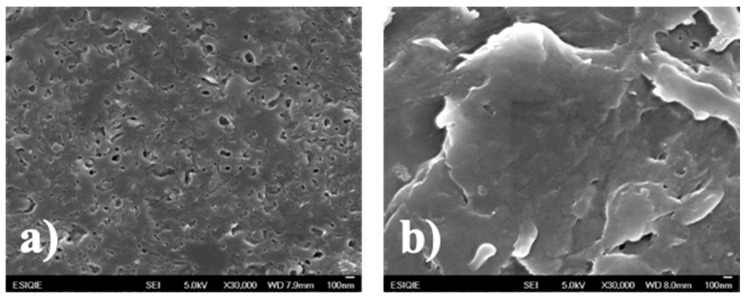
Scanning electron microscope (SEM) images of (**a**) PS/2MgZnAl-AAC (15 wt.%) and (**b**) PS/Mg2ZnAl-AAC (15 wt.%).

**Figure 7 nanomaterials-09-01528-f007:**
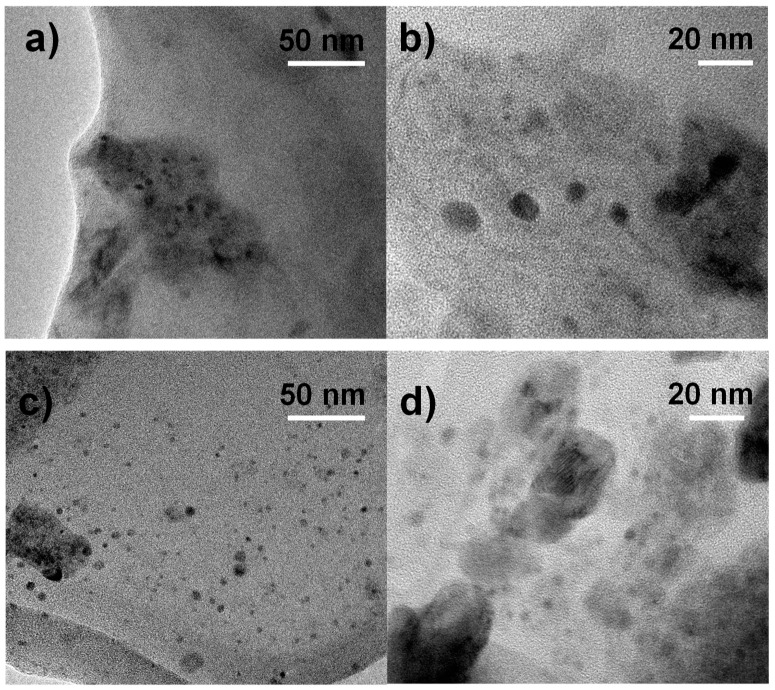
Transmission electron microscopy (TEM) images of PS/2MgZnAl-AAC (15 wt.%) (**a**) Low magnification, (**b**) High magnification, of PS/Mg2ZnAl-AAC (15 wt.%), (**c**) Low magnification image, and and (**d**) High magnification.

**Figure 8 nanomaterials-09-01528-f008:**
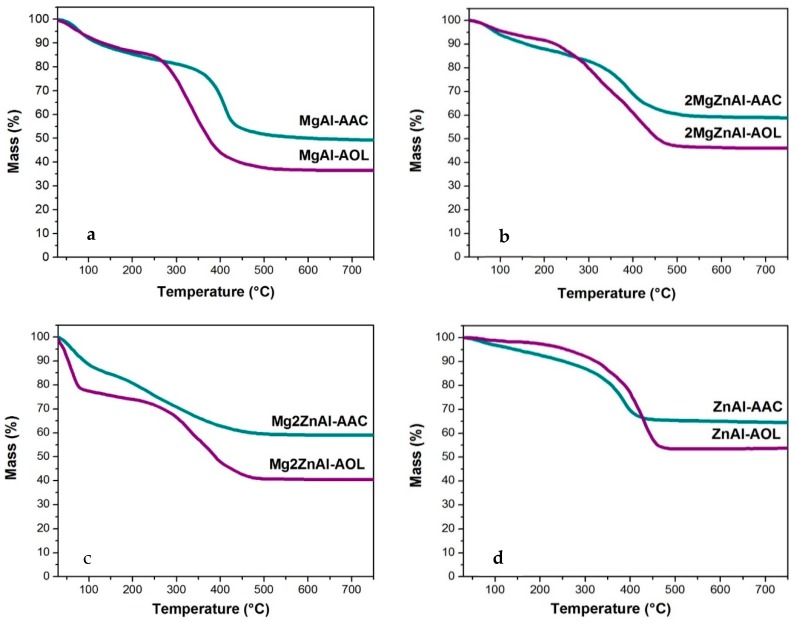
Thermogravimetric analysis for modified materials with acrylate anion (AAC) and oleate anion (AOL) for: (**a**) MgAl, (**b**) 2MgZnAl, (**c**) Mg2ZnAl, and (**d**) ZnAl.

**Figure 9 nanomaterials-09-01528-f009:**
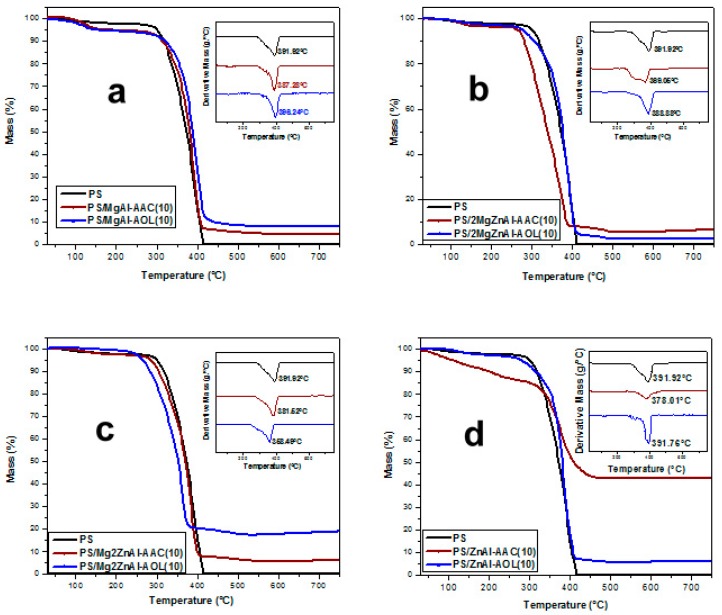
Thermograms for nanocomposites with 10 wt.% of LDH modified with acrylate (AAC) and oleate (AOL) anions for: (**a**) MgAl, (**b**) 2MgZnAl, (**c**) Mg2ZnAl, and (**d**) ZnAl.

**Figure 10 nanomaterials-09-01528-f010:**
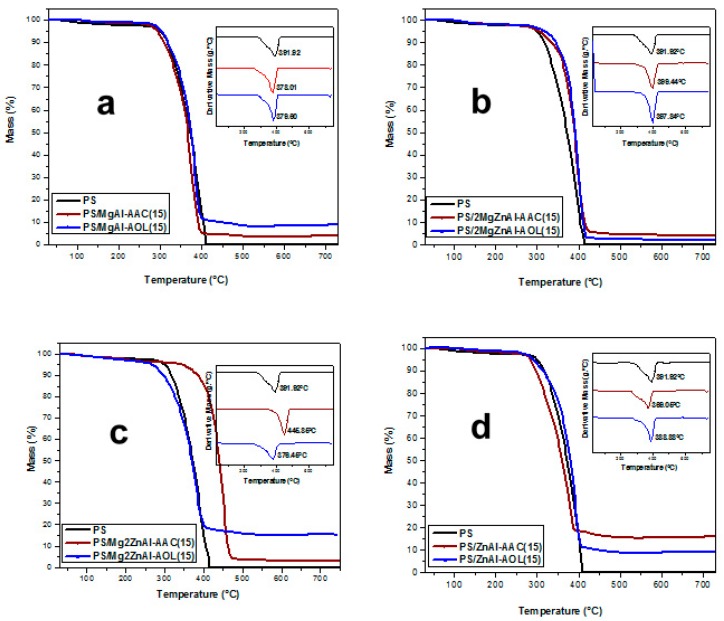
Thermogravimetric analysis for nanocomposites 15 wt.% of LDH modified with acrylate (AAC) and oleate (AOL) anions for: (**a**) MgAl, (**b**) 2MgZnAl, (**c**) Mg2ZnAl, and (**d**) ZnAl.

**Figure 11 nanomaterials-09-01528-f011:**
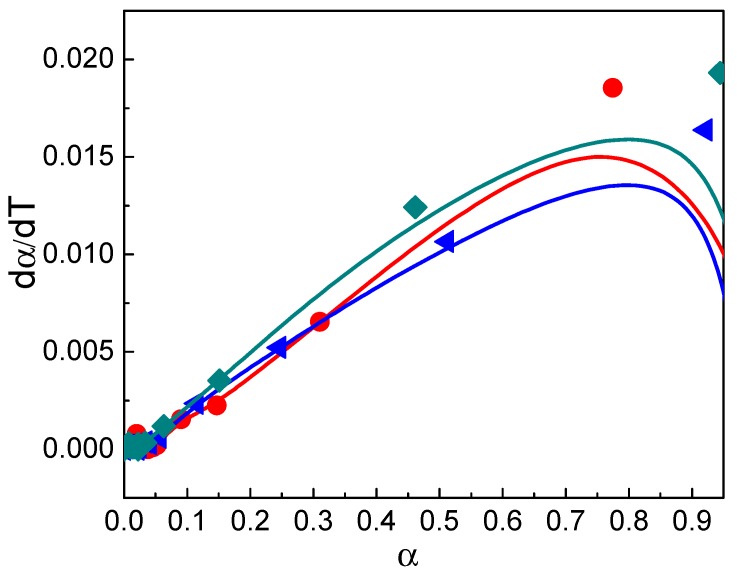
$\frac{d\alpha }{dT}$ as function of *α* for the sample PS/Mg2ZnAl-AAC (15 wt.%) at ● β = 10 °C/min, ◄ β = 20 °C/min, and ♦ β = 30 °C/min.

**Figure 12 nanomaterials-09-01528-f012:**
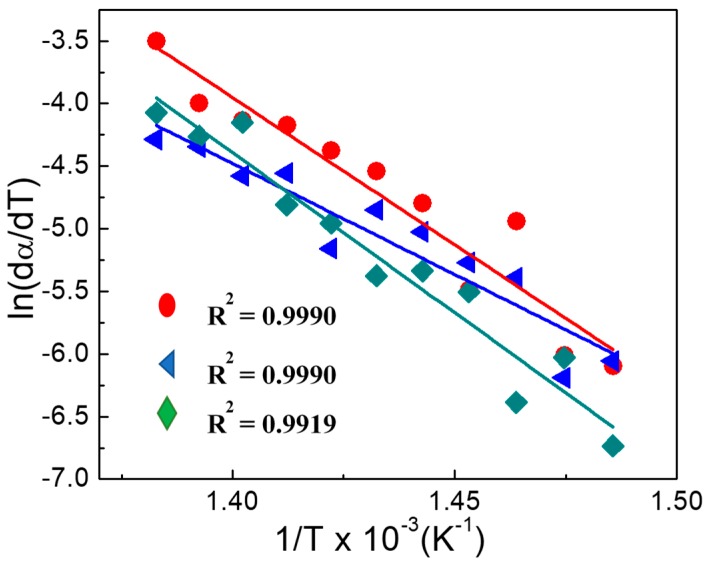
Plot of ln (*dα/dT*) vs. 1/*T* for sample PS/Mg2ZnAl_AAC (15 wt.%) at ● β = 10 °C/min, ◄ β = 20 °C/min, and ♦ β = 30 °C/min.

**Figure 13 nanomaterials-09-01528-f013:**
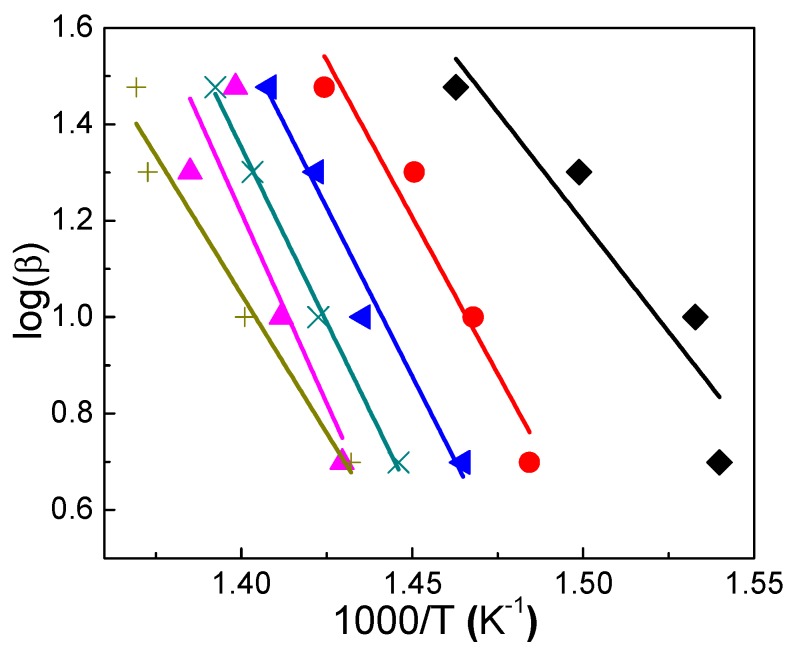
Plot of log (*β*) vs. 1000/*T* for sample PS/Mg2ZnAl-AAC (15 wt.%) at ♦ *α* = 0.1, ● *α* = 0.2, ◄ *α* = 0.3, **×**
*α* = 0.4, ▲ *α* = 0.5, and **+**
*α* = 0.6.

**Figure 14 nanomaterials-09-01528-f014:**
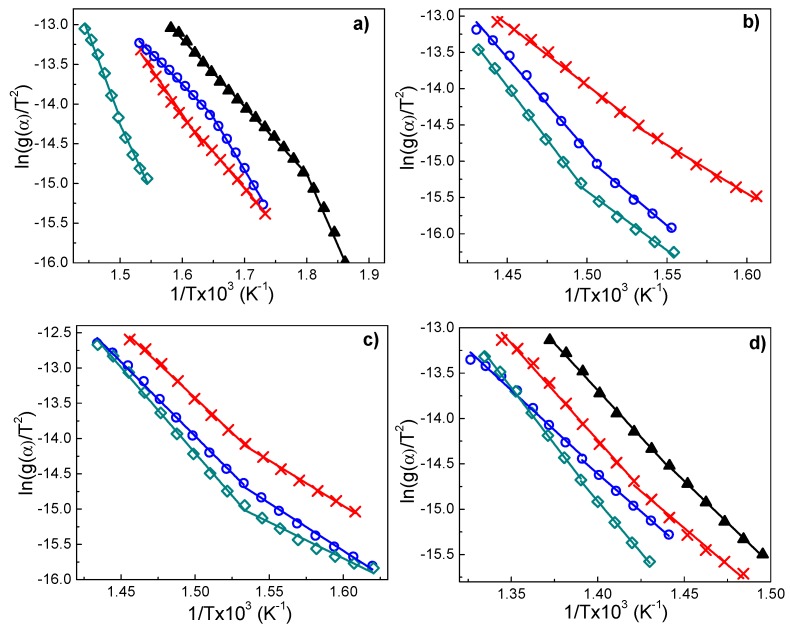
Plot of $ln\left(\frac{g\left(\alpha \right)}{T^{2}}\right)$ vs. 1/*T* for (**a**) PS/MgAl-AOL (10), (**b**) PS/2MgZnAl-AAC (15), (**c**) PS/2MgZnAl-AOL (15), (**d**) PS/Mg2ZnAl-AAC(15) at ▲ β = 5 °C/min, × β = 10 °C/min, ○ β = 20 °C/min and ◊ β = 30 °C/min.

**Table 1 nanomaterials-09-01528-t001:** Layered Double Hydroxides (LDH) materials.

LDH	Mg/Zn	Formula	Synthesis Method	AEC (mmol·g^−1^)
MgAl-COP	2/0	Mg_6_Al_2_(OH)_16_(NO_3_)_2_·4H_2_O	Co-precipitation	3.00
2MgZnAl-COP	2/1	Mg_4_Zn_2_Al_2_(OH)_16_(NO_3_)_2_·4H_2_O	Co-precipitation	2.67
Mg2ZnAl-U	1/2	Mg_2_Zn_4_Al_2_(OH)_16_(NO_3_)_2_·4H_2_O	Urea	2.40
ZnAl-U	0/2	Zn_6_Al_2_(OH)_16_(NO_3_)_2_·4H_2_O	Urea	2.19

**Table 2 nanomaterials-09-01528-t002:** Polystyrene formulation for the semicontinuous emulsion polymerization.

Reagents	Reactor (g)	Addition Tank (g)
Surfactant sol. 2.66 wt.%	3	-
Surfactant sol. 6.5 wt.%	-	30
Styrene	-	120
Na_2_S_2_O_8_ sol. 3.86 wt.%	1	10
Distilled water	37	-

**Table 3 nanomaterials-09-01528-t003:** Cell parameters of fresh and modified samples.

Sample	d_003_ (Å)	c (Å)	a (Å)
MgAl-COP	7.43	22.29	3.07
MgAl-AAC	11.62	34.85	3.05
MgAl-AOL	8.72	26.17	3.05
2MgZnAl-COP	8.73	26.17	3.05
2MgZnAl-AAC	13.30	39.91	3.03
2MgZnAl-AOL	20.27	60.81	3.05
Mg2ZnAl-U	7.58	22.75	3.06
Mg2ZnAl-AAC	12.73	38.20	3.03
Mg2ZnAl-AOL	14.40	43.90	3.05
ZnAl-U	8.73	26.18	3.07
ZnAl-AAC	15.98	47.94	2.87
ZnAl-AOL	14.40	43.19	2.95

**Table 4 nanomaterials-09-01528-t004:** Average molecular weight and polydispersity index.

${\bar{M}}_{n}\;(\mathrm{Da})$	${\bar{M}}_{w}\;(\mathrm{Da})$	*PDI*
1505.8 ± 12	1526.9 ± 13	1.01

**Table 5 nanomaterials-09-01528-t005:** Decomposition temperature of PS and nanocomposites.

Material	Decomposition Temperature (°C)	
	10 wt.% LDH	15 wt.% LDH
PS	391.9	
PS/MgAl-AAC	387.3	378.0
PS/MgAl-AOL	396.2	379.6
PS/2MgZnAl-AAC	369.1	399.4
PS/2MgZnAl-AOL	389.9	397.8
PS/Mg2ZnAl-AAC	381.5	445.9
PS/Mg2ZnAl-AOL	358.5	376.5
PS/ZnAl-AAC	378.0	369.1
PS/ZnAl-AOL	391.8	388.9

**Table 6 nanomaterials-09-01528-t006:** Activation energies obtained using the Friedman method.

Material	Activation Energy (*Ea*-F) (kJ·mol^−1^)			
	Heating Rate (°C/min)			Average
	10	20	30	
PS	50.2			50.2
PS/MgAl-AAC (10 wt.%)	103.6	86.9	102.4	97.6 ± 9
PS/MgAl-AAC (15 wt.%)	78.0	103.8	107.6	96.5 ± 16
PS/MgAl-AOL (10 wt.%)	53.9	169.4	104.1	109.1 ± 58
PS/MgAl-AOL (15 wt.%)	78.4	109.0	118.0	101.8 ± 21
PS/2MgZnAl-AAC (10 wt.%)	92.4	55.5	151.3	99.8 ± 48
PS/2MgZnAl-AAC (15 wt.%)	134.8	196.4	211.4	180.9 ± 41
PS/2MgZnAl-AOL (10 wt.%)	92.2	68.3	139.5	100.0 ± 36
PS/2MgZnAl-AOL (15 wt.%)	126.8	186.1	198.4	170.5 ± 38
PS/Mg2ZnAl-AAC (10 wt.%)	80.7	118.5	110.8	103.3 ± 20
PS/Mg2ZnAl-AAC (15 wt.%)	170.6	122.6	180.8	158.0 ± 31
PS/Mg2ZnAl-AOL (10 wt.%)	51.1	157.8	181.0	130.0 ± 69
PS/Mg2ZnAl-AOL (15 wt.%)	93.6	61.0	163.8	106.2 ± 52
PS/ZnAl-AAC (10 wt.%)	119.9	100.1	133.9	118.0 ± 17
PS/ZnAl-AAC (15 wt.%)	57.8	69.9	70.9	66.2 ± 7
PS/ZnAl-AOL (10wt.%)	105.1	107.2	109.4	107.2 ± 2
PS/ZnAl-AOL (15wt.%)	67.4	101.5	125.4	98.1 ± 29

**Table 7 nanomaterials-09-01528-t007:** Activation energies obtained using the Flynn–Wall–Ozawa method.

Material	Activation Energy (*Ea*-FWO) (kJ·mol^−1^)						
	Conversion (*α*)						Average
	0.1	0.2	0.3	0.4	0.5	0.6	
PS	50.2						50.2
PS/MgAl-AAC (10 wt.%)	83.9	64.2	74.8	118.8	113.2	120.3	95.8 ± 22
PS/MgAl-AAC (15 wt.%)	110.3	99.5	113.2	123.4	136.8	132.4	119.3 ± 13
PS/MgAl-AOL (10 wt.%)	71.4	62.3	60.8	56.7	57.7	67.9	62.8 ± 5
PS/MgAl-AOL (15 wt.%)	107.0	139.5	161.2	172.6	174.7	170.2	154.2 ± 24
PS/2MgZnAl-AAC (10 wt.%)	71.9	87.9	115.9	134.5	138.5	146.4	115.8 ± 27
PS/2MgZnAl-AAC (15 wt.%)	71.0	117.3	121.0	137.2	149.0	163.9	126.6 ± 29
PS/2MgZnAl-AOL (10 wt.%)	78.0	93.9	118.5	135.0	139.4	146.4	118.5 ± 25
PS/2MgZnAl-AOL (15 wt.%)	133.5	179.6	259.2	215.8	207.7	222.0	203.0 ± 39
PS/Mg2ZnAl-AAC (10 wt.%)	80.2	85.0	100.3	115.6	137.3	138.0	109.4 ± 23
PS/Mg2ZnAl-AAC (15 wt.%)	165.6	236.5	254.3	264.3	287.7	209.5	236.3 ± 40
PS/Mg2ZnAl-AOL (10 wt.%)	50.3	48.8	59.2	64.8	71.7	87.0	63.6 ± 13
PS/Mg2ZnAl-AOL (15 wt.%)	23.5	28.9	34.9	41.3	46.4	48.8	37.3 ± 9
PS/ZnAl-AAC (10 wt.%)	35.1	67.7	77.2	90.1	101.0	111.0	80.3 ± 25
PS/ZnAl-AAC (15 wt.%)	84.6	89.3	105.7	119.9	127.2	137.5	110.7 ± 19
PS/ZnAl-AOL (10 wt.%)	122.8	123.0	114.9	102.1	90.8	74.4	104.7 ± 17
PS/ZnAl-AOL (15 wt.%)	75.3	112.4	117.0	119.9	125.9	131.4	113.6 ± 18

**Table 8 nanomaterials-09-01528-t008:** Activation energy of the mechanism.

Material	Activation Energy (kJ·mol^−1^)		
	*Ea*-F	*Ea*-FWO	*Ea*-M
PS	-	-	50.2
PS/MgAl-AAC (10 wt.%)	97.6	95.8	96.7 ± 1
PS/MgAl-AAC (15 wt.%)	96.5	119.3	107.9 ± 11
PS/MgAl-AOL (10 wt.%)	109.1	63.3	86.2 ± 23
PS/MgAl-AOL (15 wt.%)	101.8	154.2	128.0 ± 26
PS/2MgZnAl-AAC (10 wt.%)	99.8	115.8	107.8 ± 8
PS/2MgZnAl-AAC (15 wt.%)	180.9	126.6	153.7 ± 27
PS/2MgZnAl-AOL (10 wt.%)	100.0	118.5	109.3 ± 9
PS/2MgZnAl-AOL (15 wt.%)	170.5	203.0	186.7 ± 16
PS/Mg2ZnAl-AAC (10 wt.%)	103.3	109.4	106.4 ± 3
PS/Mg2ZnAl-AAC (15 wt.%)	158.0	236.3	197.2 ± 39
PS/Mg2ZnAl-AOL (10 wt.%)	130.0	63.6	96.8 ± 33
PS/Mg2ZnAl-AOL (15 wt.%)	106.2	37.3	71.7 ± 34
PS/ZnAl-AAC (10 wt.%)	118.0	80.3	99.2 ± 19
PS/ZnAl-AAC (15 wt.%)	66.2	110.7	88.4 ± 22
PS/ZnAl-AOL (10 wt.%)	107.2	104.7	105.9 ± 1
PS/ZnAl-AOL (15 wt.%)	98.1	113.6	105.9 ± 8

**Table 9 nanomaterials-09-01528-t009:** Kinetic parameters obtained using the Coats–Redfern model.

Sample	n	Pyrolysis Process					
		First Stage			Second Stage		
		*Ea* (kJ·mol^−1^)	*A* (s^−1^)	*R* ^2^	*Ea* (kJ·mol^−1^)	*A* (s^−1^)	*R* ^2^
PS/MgAl-AOL 10 wt.%	1/3	101.8	8.4 × 10^11^	0.9926	118.7	8.5 × 10^11^	0.9919
PS/2MgZnAl-AAC 15 wt.%	1/3	198.4	2.1 × 10^17^	0.9995	133.5	1.3 × 10^10^	0.9915
PS/2MgZnAl-AOL 15 wt.%	1/3	180.2	1.8 × 10^14^	0.9866	101.5	1.5 × 10^7^	0.9810
PS/Mg2ZnAl-AAC 15 wt.%	1/3	175.8	1.5 × 10^13^	0.9900	153.6	5.7 × 10^11^	0.9375

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
