# Peer review of "Thermal Stability Evaluation of Polystyrene-Mg/Zn/Al LDH Nanocomposites"

_nanomaterials, 2019, doi:10.3390/nano9111528_

Round 1

Reviewer 1 Report

What was the reason for synthesis of PS, why not to use commercially available one?

Authors in abstract stated that low concentrations of fillers were used, but for nanocomposites 10 and 15 wt% is rather not low. Also why only 2 concentrations were analyzed? Basing on only 2 points it is hard to observe the trend, especially in nanocomposites, when after some content reversion of properties occurs.

Figure 5, spectra for PS composites, Authors are writing about some signals between 1660 and 2000, however no signals are observed in the Figure. 

Authors are writing that during TGA analysis nitrates and CO2 are released at particular temperature region. Did Authors checked it with any analysis?

Why for some samples modification with AOL gives better results on thermal stability than AAC and for other samples the opposite can be noted?

Why Authors presented only TGA results for 10 wt% of filler loading?

Author Response

Mexico City, October 21th 2019.

Dear Reviewer 1,

 We really appreciate the time you took to review our manuscript as well as your interesting comments and recommendations. The manuscript was modified according to your suggestions.

Manuscript ID: nanomaterials-619104

Title: “Thermal Stability Evaluation of Polystyrene-Mg/Zn/Al LDH Nanocomposites” by M. A. De la Rosa-Guzmán, A. Guzmán-Vargas, N. Cayetano-Castro, J. M. del Río, M. Corea, M. J. Martínez-Ortiz.

We carefully revised the manuscript based on your comments and the details of the revision that we have made are explained in the following pages.

What was the reason for synthesis of PS, why not to use commercially available one?

Because we have a polymer synthesis laboratory and for us is easier to synthesize the PS than buy it.

Authors in abstract stated that low concentrations of fillers were used, but for nanocomposites 10 and 15 wt. % is rather not low.

Compared with previous works where hydrotalcite like-materials were used as polymeric additives, authors reported contents from 2 wt. % until high contents of 60 wt.% with variable results of the thermal stability of the composites and the behavior depended of the polymer matrix used, as well as the surfactant added (J. Mat. Chem. A, 2(2014) 10996-11016).

Another work reported, specifically for polystyrene matrix with 5 wt.% of ZnAl and MgAl LDH, that decomposition temperature only increased 18°C compared with the additive-free polymer (J. Sci.: Adv. Mat. Devices, 1(2016) 351-361).

So it can be considered that 10 and 15 wt. % are low percentages if the increment of decomposition temperature for the best sample was 54°C.

Also why only 2 concentrations were analyzed? Basing on only 2 points it is hard to observe the trend, especially in nanocomposites, when after some content reversion of properties occurs.

Initially, 5, 10 and 15 wt. % loaded composites were used of the TGA experiments, nevertheless the results at 5 wt. % did not show a good behavior in thermal stability. In addition, 5 wt. % samples did not fit to the simplest Friedman model, in consequence the data of these samples would not fit the most complex models (Flynn-Wall-Ozawa and Coats-Redfern).

Figure 5, spectra for PS composites, Authors are writing about some signals between 1660 and 2000, however no signals are observed in the Figure. 

In page 8, the description of bands of PS composites was corrected (Figure 5).

Authors are writing that during TGA analysis nitrates and CO2 are released at particular temperature region. Did Authors checked it with any analysis?

 There are many reports where the thermal decomposition of the layered double hydroxides has been studied. They have established the different stages of decomposition as well as the obtained products in each one. In page 10, section 3.6, the references 47 to 49 confirm these statements. For example, the decompositions reactions for a Mg/Al LDH sample with carbonates anions is presented in reference: Thermochimica Acta 667(2018) 177-184, but references since 1975-1977 have established these decomposition stages (S. Miyata et al. Clays and Clay Miner., 23(1975) 369 and Clays and Clay Miner., 25(1977) 14).

Why for some samples modification with AOL gives better results on thermal stability than AAC and for other samples the opposite can be noted?

The aim to use two different organic anions (AAC and AOL) into the LDH samples was to hydrophobize them in order to increase their dispersion into PS matrix.

During thermal decomposition experiments, both organic anions are decomposed at relatively low temperatures, so in reality, the increase in decomposition temperature is due to the efficiency of the dispersion of the modified LDHs as well as well as their chemical composition. In our case, and for the first time for polystyrene, the use of multicationic LDH (Mg/Zn/Al) increased its degradation temperature and therefore its thermal stability.

Why Authors presented only TGA results for 10 wt% of filler loading?

Figure 8 corresponds to TGA results of modified LDH, Figure 9 presents TGA results of PS with 10 wt. % of LDH modified with acrylate (AAC) and oleate (AOL) anions and Figure 10 shows TGA results of PS with 15 wt. % of LDH modified with acrylate (AAC) and oleate (AOL) anions.

Sincerely,

Dr. María de Jesús Martínez-Ortiz

Laboratorio de Investigación en Materiales Porosos, Catálisis Ambiental y Química Fina. ESIQIE-IPN UPALM, Edif. 7 P.B. Zacatenco, 07738 Ciudad de México, Mexico.

Dr. Mónica Corea

Laboratorio de Investigación en Polímeros y Nanomateriales. UPALM, Edificio Z-5, P. B., Zacatenco, Gustavo A. Madero, C. P. 07738, Ciudad de México, Mexico.

Reviewer 2 Report

In this manuscript, the authors reported the investigation of the thermal stability of polystyrene LDH nanocomposite. The hydrophobized LDH in polystyrene were prepared and measured the decomposition temp. The authors showed the improvements for thermal stability of PSt. I recommended this work for publication in nanomaterials after major revisions.

P3, Table 1, The formula of LDH materials were shown. The differences of the structure of LDHs should be shown to understand for readers. The authors selected acrylate anion and oleate anion to hydrophobize LDHs. The reason should describe in the manuscript. The contents of acrylate anion and oleate anion in LDHs must be shown. The contents of AAC and AOL must be influenced for the thermal properties. From Figure 6 to Figure 10, the authors compared the differences of AAC and AOL containing LDHs of Pst composites. The composites of PSt-LDH before the treatment of AAC and AOL should be measured and discussed. It is necessary to clarify the effects of hydrophobic treatment of LDHs.

Author Response

Mexico City, October 21th 2019.

Dear Reviewer 2,

We really appreciate the time you took to review our manuscript as well as your interesting comments and recommendations. The manuscript was modified according to your suggestions.

 Manuscript ID: nanomaterials-619104

Title: “Thermal Stability Evaluation of Polystyrene-Mg/Zn/Al LDH Nanocomposites” by M. A. De la Rosa-Guzmán, A. Guzmán-Vargas, N. Cayetano-Castro, J. M. del Río, M. Corea, M. J. Martínez-Ortiz.

We carefully revised the manuscript based on your comments and the details of the revision that we have made are explained in the following pages.

P3, Table 1, The formula of LDH materials were shown. The differences of the structure of LDHs should be shown to understand for readers.

In fact, the LDH structure is the same for all samples, the difference between them is the chemical composition of each one. In addition, in page 3, after Table 1, the scheme of the LDH structure was incorporated in order to clarify this point (Figure 1).

The authors selected acrylate anion and oleate anion to hydrophobize LDHs. The reason should describe in the manuscript.

Acrylic and oleic acids are short-chain organic compounds. They have a good hydrophobic-hydrophilic balance; to be precise, the lipophilic part has almost the same size of the hydrophilic part. This characteristic allows that the inorganic materials as LDHs to be hydrophobized.

The contents of acrylate anion and oleate anion in LDHs must be shown.

In Page 3, Table 1, a column was added and it presents the data for anionic exchange capacity (AEC), which was used to determine the amounts of organic anions that were intercalated in the these LDH samples.

The contents of AAC and AOL must be influenced for the thermal properties.

The AAC and AOL contents do not have influence in the thermal stability of composites due to they are incorporated into the LDH materials in a constant molar ratio (see page 4, section 2.4).

From Figure 6 to Figure 10, the authors compared the differences of AAC and AOL containing LDHs of Pst composites.

In Figures 6 and 7 are presented the SEM and TEM micrographs of composites and their  morphology is discussed, based on the dispersion degree of the modified LDH into the PS matrix.

Figure 8 shows the modified LDH thermograms. In page 10 was added a paragraph where the difference of weight loss is discussed.

Figures 9 and 10 present the TGA results for the best PS-modified LDH samples with 10 and 15 wt. %, and it was discussed that the degradation temperature is influenced by the presence of modified LDH and not by the organic anions. It was observed that thermal stability was better when Zn was incorporated into the LDH.

The composites of PSt-LDH before the treatment of AAC and AOL should be measured and discussed.

The chemical nature of inorganic LDH and PS (organic polymer), it makes not possible the LDH incorporation into the polymer. If these experiments were carried out without the organic anionic species (AAC and AOL), there would be a segregation of phases between the inorganic and polymer materials.

It is necessary to clarify the effects of hydrophobic treatment of LDHs. 

This point was treated at the end of the introduction section.

Sincerely,

Dr. María de Jesús Martínez-Ortiz

Laboratorio de Investigación en Materiales Porosos, Catálisis Ambiental y Química Fina. ESIQIE-IPN UPALM, Edif. 7 P.B. Zacatenco, 07738 Ciudad de México, Mexico.

Dr. Mónica Corea

Laboratorio de Investigación en Polímeros y Nanomateriales. UPALM, Edificio Z-5, P. B., Zacatenco, Gustavo A. Madero, C. P. 07738, Ciudad de México, Mexico.

Reviewer 3 Report

Dear Authors, 

After reading the document "Thermal Stability Evaluation of Polystyrene-Mg/Zn/Al LDH Nanocomposites" this reader acknowledges both, the ammount of experimental data collected and the potential relevance of some of the reported results. In spite of this positive impression, the document you present is not optimal according to this reviewer. In general, the analysis  and description of the results is expeditive and there are plenty of relevant aspects that should be improved/highlighted. Paradoxically, a lot of information of marginal interest is integrated in the article, which avoids fixing the relevance of the main contributions. The document should thus be drastically condensed. Additionally, the statistical relevance of the most promising results should be justified. You can find below a list of the most prominent criticisms. Please consult the annotated version to illustrate where this criticisms emerge and some additional minor comments/suggestions. 

The document needs a moderate revision of the use of English. Note already initial mistake in the abstract. A revision of most of these errors has been integrated in the annotated version, but conforming to these suggestions may not be enough to make the article fully satisfactory in this aspect.  Please follow instructions to improve the introduction. Is there a particular reason why your materials could improve previous systems as flame retardants? Note unconventional concluding remarks at the end of the introduction. In the experimental section, you will have to justify the need of the annealing treatments. Are them just to justify the byproducts of the degradation? Also, comment on how did you proceed to evaporate toluene... consider that your process should be integrally environmentally friendly.  Is figure 1 strictly necessary taking into account that your synthesis is based in previous processes?   In XRD of synthesised products there are peaks that have to be labelled (especially ZnAl-U sample). For the annealed samples, since the analysis is relevant only for analytical purposes and not so much for functional purposes, you should present these diffraction data with the TGM analysis.  Reshape figure 3 of diffractograms for intercalated LDHs... why is there segregation of ZnO in the sample with less Zn? Or, why is the sample with more Zn providing the best results of intercalated LDH structures? Since there is not an actual study of AcAc and OLA conformation upon intercalation, your mechanisms of intercalation appear as speculative. Instead of "corroborate" use "suggest" when you refer to external results to justify your observations.  For FTIR results, include labels of the main molecular species responsible for the different modes. Please comment on the main differences of the analysis on alternative samples to those presented. For TEM , note images c and d) are not mentioned in the text. Could the observed planes be related to some of the structures detected by XRD... Please make a big effort to show strictly the most relevant results from the point of view of TGA. Show those for the most relevant sample (Mg2ZnAl) and comment relevant aspects for the rest. You should envisage to provide a single figure with the c) plot of the current figs 8-10. Table 5 is illustrative enough to compare the performance of the different samples.  With respect to the analysis of the kinetics, are all three models relevant? Is there one outstanding over the others? Which is best to fir the results of the most relevant sample? According to your reply, present results according to one single model (not necessarily for all the samples) and a summarizing table. Paradoxically, so many data are detrimental for the main impression of your work.  With regards to the comment "Thus, activation energy variations indicate that thermal decomposition of the nanocomposites depends on good dispersion of the additive in the polymeric matrix, as well as the concentration of additive used in the formulation" be sure to compare with data referring to non intercalated composites. Otherwise, the comment is speculative. Additionally, you should make an effort to explain why there is such a critical effect of an increase of LDH concentration. This reviewer is forced to ask if this result is representative... How many times was this experiment performed?

Yours

Author Response

Mexico City, October 21th 2019.

Dear Reviewer 3,

We really appreciate the time you took to review our manuscript as well as your interesting comments and recommendations. The manuscript was modified according to your suggestions.

 Manuscript ID: nanomaterials-619104

Title: “Thermal Stability Evaluation of Polystyrene-Mg/Zn/Al LDH Nanocomposites” by M. A. De la Rosa-Guzmán, A. Guzmán-Vargas, N. Cayetano-Castro, J. M. del Río, M. Corea, M. J. Martínez-Ortiz.

We carefully revised the manuscript based on your comments and the details of the revision that we have made are explained in the following pages.

Is there a particular reason why your materials could improve previous systems as flame retardants?

The principal goal of this work is that we present for the first time results of the use of multicationic Mg/Zn/Al LDH modified with acrylate and oleate anions at low contents (10 and 15 wt. %) into PS, increasing the thermal stability about 57 °C in the best sample. There are many previous works related to the incorporation of single LDH materials where the contents reported were higher than 10 wt. % with modest results.

In the introduction section was added the following paragraph: “It has been reported that LDH materials as flame retardant, present three characteristics through the thermal decomposition as heat absorption, gas dilution and ash formation [5]. In addition, during this process, the loss of interlayer water in LDH, as well as intercalated anions and the water generated by the dehydroxylation, allowing the reduction of the available fuel. For this reason, the mass loss rate is significantly reduced”.

Note unconventional concluding remarks at the end of the introduction.

The text was modifies according to the request.

In the experimental section, you will have to justify the need of the annealing treatments. Are them just to justify the byproducts of the degradation?

In section 2.3 is exposed the procedure of the LDH thermal treatment before the intercalation of AAC and AOL anionic species. This step is necessary to obtain the mixed oxides of the Mg/Zn/Al inorganic layered materials.

One of the most important properties of LDH materials is the “memory effect”. This property is achieved by calcining the fresh LDH at temperatures not higher than 500 °C. Then, structures are regenerated to the initial LDH in aqueous medium. This must be carry out with the purpose of introduce new anionic species, different of the initial ones (NO3- in this case), because the NO3- anions have a greater affinity in the LDH structures [F. Cavani, F. Trifirò, A. Vaccari, Catal. Today, 11(1991) 173-301], thus a direct anionic exchange in fresh materials would not allow the incorporation of AAC and AOL organic anions.

Also, comment on how did you proceed to evaporate toluene... consider that your process should be integrally environmentally friendly.

The solvent was removed and recuperated with a rotavapor at room temperature.

This statement was included in the experimental section 2.6.

Is figure 1 strictly necessary taking into account that your synthesis is based in previous processes?  

Figure 1 related to the polymer synthesis was removed from the manuscript.

In XRD of synthesised products there are peaks that have to be labelled (especially ZnAl-U sample). For the annealed samples, since the analysis is relevant only for analytical purposes and not so much for functional purposes, you should present these diffraction data with the TGM analysis. 

In Figure 2, all diffractograms are labelled for their corresponding crystalline phases. In the case of Figure 2a, all samples present the characteristic hydrotalcite phase (JCPDS card: 22-0700). Figure 2b corresponds to samples that exhibited comparable XRD patterns corresponding to the mixed oxides Mg(Al)O structure, analogous to periclase-like structure MgO (JCPDS card 04-0829) or/and ZnO (JCPDS card: 00-036-1451), see section 3.1. In addition, the indexation of planes were added to the diffractograms.

With respect to the XRD data analyzed by TGM, we are confuse about your request. If you refer to follow the crystalline structure of materials as a function of temperature (thermodiffraction) or a refinement method, we do not have the technique or software, but all diffractograms are referred to the crystallographic cards.

Reshape figure 3 of diffractograms for intercalated LDHs...

Figure 3 was modified.

why is there segregation of ZnO in the sample with less Zn?

In section 2.3 it was described that samples prepared by coprecipitation method were calcined in air flow, this caused the segregation of ZnO crystalline phase.

Or, why is the sample with more Zn providing the best results of intercalated LDH structures?

Section 3.1. The results of DRX showed that this sample presents a pure LDH phase and its crystalline degree is higher than the rest of materials. Moreover, in section 3.5 is discussed that this sample presents homogeneous and well dispersed LDH particles into the PS matrix. Thus, both results can be associated to the good performance of this sample as flame retardant of PS.

Since there is not an actual study of AcAc and OLA conformation upon intercalation, your mechanisms of intercalation appear as speculative. Instead of "corroborate" use "suggest" when you refer to external results to justify your observations. 

 The text was modified.

For FTIR results, include labels of the main molecular species responsible for the different modes.

Figure 5 was modified.

Please comment on the main differences of the analysis on alternative samples to those presented.

 This point was explained in the introduction section where several previous works using PS were mentioned in order to remark their main results and achievements.

For TEM, note images c and d) are not mentioned in the text. Could the observed planes be related to some of the structures detected by XRD...

The text was modified and the figures c and d were mentioned in the text as you indicate.

Certainly, as you comment, the planes can eventually be related to the principal planes of LDH structure, but it was not the main propose in the present work. This study was mainly focused on the thermal stability of the composite materials.

Please make a big effort to show strictly the most relevant results from the point of view of TGA. Show those for the most relevant sample (Mg2ZnAl) and comment relevant aspects for the rest.

Based on the comments of the other reviewers, we considered the most of your suggestions and we try to take into account all of your valuable observations in order to depict our main results.

You should envisage to provide a single figure with the c) plot of the current figs 8-10.

We really appreciate your suggestion and it could indeed be clearer if we focused only on the best sample, however we consider it is important to show the results of the other samples with the idea of having a more complete picture of the thermal behavior of this family of materials.

Table 5 is illustrative enough to compare the performance of the different samples.  With respect to the analysis of the kinetics, are all three models relevant? Is there one outstanding over the others?

The importance of Table 5 was the election of the four best samples to develop the thermal stability studies.

In the conclusions section was exposed this point as: “The obtained Ea by the Friedman and Flynn-Wall-Ozawa methods presents different values attributed to a two stages decomposition process. This was confirmed by the results of Coats-Redfern method where in the first stage, Ea was higher than the second stage assuming that this behavior was due to the presence of the modified LHD into the polymeric matrix. From this model, a reaction order of 1/3 was obtained; this indicates that during the decomposition process, a recombination of free radicals takes place due to the addition of LDH materials modified with organic short chain compounds”.

Which is best to fir the results of the most relevant sample? According to your reply, present results according to one single model (not necessarily for all the samples) and a summarizing table. Paradoxically, so many data are detrimental for the main impression of your work.

We consider that the best model to fit the data of these composite materials was the Coats-Redfern method because it represents well the multi-stages decomposition processes.

With regards to the comment "Thus, activation energy variations indicate that thermal decomposition of the nanocomposites depends on good dispersion of the additive in the polymeric matrix, as well as the concentration of additive used in the formulation" be sure to compare with data referring to non intercalated composites. Otherwise, the comment is speculative.

The chemical nature of inorganic LDH and PS (organic polymer), it makes not possible the LDH incorporation into the polymer. If these experiments were carried out without the organic anionic species (AAC and AOL), there would be a segregation of phases between the inorganic and polymer materials.

Additionally, you should make an effort to explain why there is such a critical effect of an increase of LDH concentration. This reviewer is forced to ask if this result is representative... How many times was this experiment performed?

For the best sample, it is multicationic Mg/Zn/Al LDH, this sample presented a pure hydrotalcite-like material crystalline phase without an additional phase like the rest of the samples containing Zn. This caused that the modified LDH material was better dispersed into de polymeric matrix. When the thermal decomposition process proceeded, these materials generate and release more water helping to decrease the polymer decomposition.

It is important to mention that all experiments were carried out by triplicate and all results have a standard deviation associated. For this reason, we consider that the obtained results are representative of phenomena of thermal decomposition.

Finally, all request and observations mentioned in the PDF file were attended.

Sincerely,

Dr. María de Jesús Martínez-Ortiz

Laboratorio de Investigación en Materiales Porosos, Catálisis Ambiental y Química Fina. ESIQIE-IPN UPALM, Edif. 7 P.B. Zacatenco, 07738 Ciudad de México, Mexico.

Dr. Mónica Corea

Laboratorio de Investigación en Polímeros y Nanomateriales. UPALM, Edificio Z-5, P. B., Zacatenco, Gustavo A. Madero, C. P. 07738, Ciudad de México, Mexico.

Round 2

Reviewer 2 Report

This paper has been successfully revised. 

Reviewer 3 Report

Dear Authors, 

You have complied to most of the criticisms raised by this reviewer. Although I personally do not agree with the inclusion of some extensive publication, your article has notably increased in quality. 

Note some LHD acronym in page 13 (instead of LDH). 

Yours,